# A Taxonomy of Agentic Errors: A systematic review of how agentic AI systems can fail

## Abstract

Agentic AI extends generative AI by enabling AI systems to select and take actions in the physical or digital world. With new capabilities and greater autonomy, new errors can occur in agentic AI systems. Addressing these errors requires a more comprehensive understanding of the types of possible errors. To address this gap, we performed a systematic literature review of agentic AI errors. We surveyed 1,379 agentic AI research papers. Of those, we identified 123 papers that discussed errors that occurred in their agentic AI systems. Then, we extracted 652 errors which we qualitatively classified into a formal taxonomy. Our goal is to provide a foundational understanding of agentic AI errors to enable a more systematic approach to identifying and addressing these errors.

## 1. Introduction

Agentic Artificial Intelligence (AI) extends generative AI capabilities with the ability to reason and act. These new capabilities enable AI agents to interact with resources, services, and tools to perform complex tasks with a degree of autonomy and limited supervision (Acharya et al., 2025; Miehling et al., 2025). When working as expected, agentic AI systems are useful tools to increase productivity. However, when these complex systems generate unexpected output or fail to execute their requested task correctly, it can be challenging to understand why (Shavit et al., 2023). In addition to errors caused by the generative AI powering the agent or errors stemming from tools used by the agent, errors can also be attributed to reasons unique to agentic AI (Sapkota et al., 2026).

Understanding these errors, the causes of an agent's failure to complete a task, is fundamental to the development and design of robust agentic AI systems. Establishing a systematic understanding of agentic AI errors would shift the approach to addressing such errors from a reactive, piecemeal process to a proactive and holistic one. Additionally, frameworks and tools for building and evaluating agentic

AI systems can leverage this understanding in their design to encourage better practices during development, resulting in less error-prone agents.

To advance this vision of more robust agentic AI, we conducted a systematic literature review of agentic AI research published over the last five years to catalog the ways in which agentic systems fail. From an initial set of 1,397 results, we narrowed down the selection to 123 relevant papers and extracted 652 agentic AI errors that were mentioned within them. We then qualitatively analyzed and clustered these errors, grouping similar failure modes together to organize our findings into the first comprehensive taxonomy of agentic AI errors. Our primary contribution is a taxonomy of 72 types of agentic AI errors grounded in a systematic review of prior work. This taxonomy offers a shared reference that can support future work on the development and assessment of agents, considering the cascade effects of agents' errors in agent design, and developing more reliable ways of identifying and mitigating errors.

## 2. Related Work

To properly ground our agentic AI error taxonomy, we must first define what we mean by *agentic AI*. As definitions of the term vary, we set our definition based on prior established work. We also build atop earlier work that has already identified a subset of agentic AI errors.

### 2.1. Agentic AI and AI agents

Previous research describes agentic AI systems in a variety of ways. For example, Acharya et al. (2025) posits that "Agentic AI includes the class of autonomous AI systems that undertake to finish a set of complex tasks that span over long periods of time without human supervision." Miehling et al. (2025) describes an agentic AI system as "a collection of agents interacting with humans and the environment with the objective of fulfilling specified goals." What most descriptions have in common are a set of characteristics: autonomy, ability to break down complex tasks, and ability to interact with the external environment. Most definitions of agentic systems contain some notion of independence, or ability to take actions to complete tasks with minimal

or no human direction (Shavit et al., 2023; Miehling et al., 2025). This independence means that humans can provide fewer specific instructions, and agentic systems can handle this "underspecification" by obtaining context or requesting clarification (Chan et al., 2023). Agentic systems are typically able to handle more complex and ambiguous tasks than other kinds of AI systems, such as longer-term tasks or changing tasks that require flexibility or adaptation (Shavit et al., 2023; Chan et al., 2023; Miehling et al., 2025). Finally, agentic systems are typically able to interact with the external environment (Shavit et al., 2023). For example, they may call an API to gather information for a task or to take an action, such as sending an email (Miehling et al., 2025). We leverage existing work defining agentic systems to support our paper selection decisions.

### 2.2. Taxonomies and Agentic AI Failures

Given the breadth of AI research, the topic of how AI systems fail has been viewed through numerous lenses. One common approach is to narrow in on a subset of errors such as those for autonomous vehicles (Joshi & Kumar, 2024), speech generators (Hutiri et al., 2024), or AI functionality failures (Raji et al., 2022). AI failures have also been considered from more holistic perspectives. For example, Zhan et al. identified six "failure modes," which they define as "the manner in which a breakdown is perceived to occur at the intersection of the technical components, human agents, and actions surrounding an AI artifact" (Zhan et al., 2023). However, as identified in prior work, *comprehensive taxonomies* of failures in agentic AI systems are unfortunately lacking (Pan et al., 2025). According to the IEEE Standard Glossary of Software Engineering Terminology, a taxonomy is "a scheme that partitions a body of knowledge and defines the relationships among the pieces" (IEE, 1990). By structuring and relating different types of risks, taxonomies support prioritization and enable stakeholders to group, compare, and organize risks by type, severity, and context. In doing so, they provide a foundation for more rigorous and transparent risk analysis and mitigation.

In recent years, a growing number of taxonomies have been developed to classify risk categories in emergent technologies, often revealing risks that were previously under-recognized or poorly articulated. As technologies continue to evolve, the development of such taxonomies becomes increasingly important as a systematic way to organize and potential failures. More commonly found and closely related to errors are taxonomies of AI risks. These taxonomies aim to catalog how AI systems can cause harms to enable understanding and mitigation of such harms (Tabassi, 2023; Slattery et al., 2024; Bagehorn et al., 2025; Shelby et al., 2023; Weidinger et al., 2022).

A distinctive characteristic of agentic AI failure modes is

the fact that they result from interactions among multiple autonomous agents operating in shared and dynamic environments, rather than emerging from a single model (Sapkota et al., 2026). This interaction between agents increases the possibilities of error cascades, where the behavior of one agent alters the operational and decision context for others. Furthermore, previous research has highlighted that as agentic AI systems grow in the number of agents involved and the diversity of their specialized roles, tracing the root cause of a failure becomes increasingly complex because it requires disentangling "nested sequences of agent interactions, tool invocations, and memory updates, making debugging non-trivial and time-consuming" (Sapkota et al., 2026).

In order to develop more robust evaluation approaches for agentic systems, recent research efforts have proposed frameworks and taxonomies to identify agentic systems' errors and facilitate their debugging and mitigation. For instance, Kartik et al. (2025) introduce AgentCompass, an evaluation framework tailored for post-deployment monitoring and debugging of agentic workflows. As part of this framework, the authors proposed a "hierarchical error taxonomy" intended to be used as the guiding schema for agents' reasoning. Examples of formalized taxonomies include a taxonomy of errors developed by Winston & Just (2025), based on two "tool-augmented LLMs," identifying a total of 7 root causes of failure. Cemri et al. focused on multi-agent systems and identified 14 types of failure modes spanning pre-execution, execution, and post-execution (Pan et al., 2025).

Another line of work develops taxonomies in conjunction with benchmark design for evaluating autonomous agents on representative tasks. Lu et al. (2025b), for example, developed a three-tier taxonomy of failure causes after an in-depth failure analysis of the results of benchmarking 34 representative tasks designed to assess autonomous agents. Similarly, Deshpande et al. (2025) formalize a taxonomy of agentic error categories spanning reasoning, planning, and execution, as part of the development of the TRAIL benchmark.

Unlike existing taxonomies and frameworks, which often focus on a subset of agentic AI systems or are developed as part of benchmarks or monitoring tools, the taxonomy we propose is comprehensive and derived from a systematic literure review of 123 agentic AI papers, in which practitioners themselves, and hence the research community, determine what constitutes an error in the development of agentic systems. With this work, we aim to provide a foundational step towards developing a more systematic and holistic approach to addressing emerging errors in agentic AI systems.

## 3. Methods

We use PRISMA (Preferred Reporting Items for Systematic reviews and Meta-Analyses) to report our systematic review methods (Page et al., 2021). We performed a search, filtered the results, extracted errors, and grouped the errors into a taxonomy.

Before beginning our formal search, we experimented with several scholarly search platforms (Scopus, Google Scholar) and various search terms. We decided to use Google Scholar, as it provided a wide breadth of venues and the best flexibility in how we could perform the searches. Notably, Google Scholar indexed the full content of the paper, which was helpful because errors were often unmentioned in paper abstracts. Our aim was to find papers that included agentic AI systems, so we selected a set of terms often used to describe such systems: "agentic," "llm agent," "ai agent," and "generative agent." We also chose a set of terms typically used to describe instances when these systems misbehaved or stopped working, such as "error analysis" or "failure type" (see all terms in Table 1). We chose these terms to cast a broad net, aiming to capture as many papers as possible describing types of agentic errors.

Once we chose our search platform and terms, we performed two searches. The first search was in April 2025, which encompassed all publication venues. We performed a second search in August 2025 to capture papers published in key venues with 2025 publication dates after our original search. These venues included ACL (July 2025), ICLR (April 2025) and ICML (July 2025). Our two searches returned 1,397 results.

To filter the results, five researchers divided the search results, independently and manually determining their eligibility for our study. We had three core inclusion criteria:

1. Papers needed to be published in a peer-reviewed venue (including workshops, but not including books, dissertation, tutorials, or magazine articles).

2. Papers needed to be about an agentic AI system (based on our current understanding and definition of these systems). To be considered agentic, we required the paper to discuss a system which used generative AI to decide which actions to take and execute those actions by interacting with systems, virtual environments, or environments external to the generative model.

3. Papers needed to include a description of errors or failures of that system. We allowed all errors or failures as described in the papers rather than using our own definition of error or failure.

When the eligibility of a paper was unclear, multiple researchers discussed the paper together. In total, we kept 123 papers.

After filtering, the researchers again divided and independently extracted errors from the papers into a spreadsheet. This involved a second check of the papers to ensure that they had the required agentic errors. Again, the authors discussed all cases where ambiguities or questions about the error extraction occured. In total, we extracted 652 errors from the 123 papers.

We then moved the extracted errors to Mural, an online visual collaboration tool, to enable qualitative clustering and analysis. The five researchers worked together to group the extracted errors. At least two researchers checked each emerging group and discussed placement of each error in its group, moving it or modifying the groups when necessary. After confirming groupings, we formalized them into named categories, discussing each category with the team. The final taxonomy includes error categories and errors.

## 4. Results

Our constructed taxonomy has 11 agentic AI error categories with 72 total errors, spanning errors pertaining to the input or instructions, planning errors, tool errors, reasoning errors, and errors in execution and outputs. Table 2 shows the definitions of each category and counts of each error. More detailed information is available in Table 3 in the Appendix. In this section, we detail the types of errors and provide examples to shed light on how these errors manifest.

### 4.1. Instruction Understanding Errors

*Instruction understanding errors* occur when the LLM responsible for the agent's behavior fails to or is unable to follow instructions correctly. Instruction handling is especially important in agentic systems, as agentic systems can ideally handle complex and ambiguous tasks. Most frequently, these errors occurred due to the LLM either failing to understand and execute the specified instructions, or not completely fulfilling the specifications or requirements given in the prompt. These error manifested as prompt misunderstandings (Moncada-Ramirez et al., 2025), failures to understand an end user's questions (Rivkin et al., 2024), and ignoring task constraints (Wu et al., 2025; Pan et al., 2025). Papers also reported that such errors occurred due to inadequate or ambiguous instructions provided by the agent's end user. Prompts that supply instructions to the LLM may also include additional task context. In some instances, errors emerged from misunderstandings of this task context. For example, an agent may fail to account for environmental factors in a code generation task (Huang et al., 2024c).

*Table 1.* Initial Google Scholar searches. All searches included the must include term AND at least one of "error analysis" "failure analysis" "error type" "failure type" "error taxonomy" "failure taxonomy" "error cause" "failure cause" "error mode" "failure mode" "error evaluation" "failure evaluation".

| Must include | Publication | Publication Years | Search Date | # of Results |
| --- | --- | --- | --- | --- |
| "agentic" | All | 2022-2025 | 4/8/25 | 469 |
| "llm agent" | All | 2022-2025 | 4/9/25 | 361 |
| "ai agent" | All | 2022-2025 | 4/17/25 | 381 |
| "generative agent" | All | 2022-2025 | 4/17/25 | 43 |
| "agentic" | ACL, ICLR, ICML | 2025 | 8/25/25 | 15 |
| "llm agent" | ACL, ICLR, ICML | 2025 | 8/25/25 | 51 |
| "ai agent" | ACL, ICLR, ICML | 2025 | 8/25/25 | 49 |
| "generative agent" | ACL, ICLR, ICML | 2025 | 8/25/25 | 28 |

## 4.2. Input and Information Processing Errors

The *input and information processing error* category captures errors arising from problems with non-prompt inputs and issues related to the LLMs processing that information. This category of errors is especially salient to agentic systems as they are often framed as systems that can collect and process data of different types and sources. The most commonly reported error was the agent selecting the wrong input for its assigned task, either by selecting the wrong resource (Cai et al., 2025), or failing to extract the relevant information from an input (Chen et al., 2024b; Jiao et al., 2024). Similar to how LLMs can misunderstand the prompt, LLMs can also misunderstand other kinds of input supplied by the agent. Errors could be caused by data format or syntax issues (Wu et al., 2025; Brown et al., 2024) and by mistakes in understanding visual inputs (Han et al., 2024). For information processing, papers reported that agents struggled with interpreting three-dimensional spaces (Chu et al., 2024; Kim et al., 2024), text comprehension (Huang & Xiao, 2024), and performing arithmetic (Khandekar et al., 2024; Crispino et al., 2024).

## 4.3. Knowledge Gaps

*Knowledge gap* errors occur when agents have insufficient training or context for the given task. The most common version of this error was reported as a lack of domain knowledge— lacking task-specific knowledge needed to execute a task correctly. Existing "common sense" knowledge used to train the underlying LLM also caused errors. Accordingly, some papers described agent errors as a "lack of common sense." Examples include trying to filter web page results using the search bar instead of the filter control (Yu et al., 2025), and confusing a light-hearted statement from a speaker as a serious one (Maharana et al., 2024).

*Knowledge gaps* about context were grouped into two categories. Context-related knowledge gaps are more related to the agentic qualities of systems rather than purely the LLM,

as agentic systems are often making decisions about which context to consider or how to keep track of the context. One such error was due to missing context. In these situations, necessary context was either never provided to the agent (Butala et al., 2024), or context was never correctly created by the agent itself over the course of the task (Wang et al., 2024c). Papers also reported on how agents lost context over the course of the task (Brown et al., 2024; Pan et al., 2025). These errors were especially salient in longer-term tasks in which agentic systems lost context over time.

## 4.4. Reasoning Errors

*Reasoning errors* occur when the agent is trying to solve a given task. This category of errors captures ways that an agent fails to reason correctly about information. In some cases, these reasoning errors are related to specific qualities of agentic systems, like integrating information from multiple sources (Huang et al., 2024a) or reasoning over multiple steps of a task (Matlin et al., 2025a). Additional reasoning errors include problems specializing or generalizing information (Wu et al., 2025), having too much complexity in terms of number of entities to reason about (Sravanthi et al., 2025), or having too many steps (Wu et al., 2025). Agents also struggled with causality (Maharana et al., 2024; Matlin et al., 2025a), or their reasoning was contaminated by the underlying model's training data (Matlin et al., 2025a).

## 4.5. Incorrect Planning

*Incorrect planning* refers to a collection of failures where an agent generates a plan, or a set of actions to take, that is flawed, incomplete, or misaligned with the task's goals. Planning is a critical capability for agentic systems, as a key feature of an agentic system is the ability to select the set of actions to take. *Incorrect planning* failures emerge during the process of planning in four different ways. First, when agents lack a deliberate plan: instead of reasoning through the task and decomposing a problem, an agent relies on memorized examples (Caccia et al., 2024), uses greedy or brute-force approaches (Islam et al., 2024a), or skips rea-

*Table 2.* The Agentic AI Error Taxonomy. Errors are grouped in Error Categories along with counts of each error across all papers.

| Error Category | Definition | Error | # of errors |
|---|---|---|---|
| Instruction Understanding Errors | The agent does not follow instructions, typically provided by a system or end-user prompt. | LLM fails to understand instructions | 33 errors |
| | | Prompt or instructions are ambiguous | 5 errors |
| | | LLM fails to understand task context | 3 errors |
| Input and Information Processing Errors | The agent does not handle non-prompt inputs such as data stores, applications, resources, or other agents correctly. | Incorrect input retrieval | 22 errors |
| | | Visual input processing error | 13 errors |
| | | Problems with arithmetic and numbers | 9 errors |
| | | Input formatting/syntax errors | 7 errors |
| | | Text comprehension error | 7 errors |
| | | LLMs don't understand 3D spaces | 6 errors |
| Knowledge Gaps | The agent makes mistakes due to its knowledge or its task approach. | Lack of domain knowledge | 13 errors |
| | | Missing context | 9 errors |
| | | Context loss over time | 9 errors |
| | | Lack of common sense | 7 errors |
| | | LLM training data affects correctness | 6 errors |
| Reasoning Errors | The agent makes mistakes due to its knowledge or its task approach. | Unspecified reasoning errors | 6 errors |
| | | Information integration reasoning problems | 5 errors |
| | | Problems generalizing or specializing knowledge | 5 errors |
| | | Causal reasoning errors | 3 errors |
| | | Task complexity too high | 2 errors |
| | | Training data biases reasoning | 1 error |
| Incorrect Planning | The agent does not correctly plan its execution of the task. | Plan does not match stated goals | 16 errors |
| | | Missing steps/actions in plan | 9 errors |
| | | Unspecified planning error | 7 errors |
| | | Unfollowable action plan | 6 errors |
| | | Lack of a deliberate plan | 5 errors |
| | | Difficulty planning over a long time | 2 errors |
| | | False assumptions in plan | 2 errors |
| | | Syntax error in plan | 2 errors |
| Non-optimal Execution | The agents executes its task with extra, unnecessary actions. | Repetition in actions | 20 errors |
| | | Incorrect steps/actions in plan | 15 errors |
| | | Action budget limits | 11 errors |
| | | Unspecified non-optimal execution | 9 errors |
| | | Inadequate verification | 4 errors |
| | | Extra actions | 1 error |
| Incorrect Execution | The agent does not complete its task or does not complete it correctly. | Unspecified incorrect actions | 36 errors |
| | | Failure to complete plan execution | 15 errors |
| | | Invalid actions | 12 errors |
| | | Missing steps/actions in plan execution | 10 errors |
| | | Incorrect UI interaction | 9 errors |
| | | Cross-agent communication failures | 6 errors |
| | | Propagation of errors | 6 errors |
| | | Unspecified execution error | 5 errors |
| | | Agent refusal to execute | 4 errors |
| | | Executing non-existent actions | 3 errors |
| | | False assumptions in execution | 2 errors |
| | | Doesn't ask for clarification | 2 errors |
| Tool Errors | The agent uses tools incorrectly or tool problems prevent task completion. | Unable to navigate websites | 15 errors |
| | | Tool is incorrect for the desired action | 10 errors |
| | | Tool is called with incorrect parameters | 10 errors |
| | | LLM context length and input size limitations | 9 errors |
| | | Tool throws an error | 8 errors |
| | | Tool not applicable for current context | 6 errors |
| | | Tool is called with incorrect syntax | 5 errors |
| | | Other types of tool errors | 4 errors |
| | | Tool is called with incorrect configuration | 2 errors |
| | | Tool does not exist | 2 errors |

| Error Category | Definition | Error | # of errors |
|---|---|---|---|
| Hallucination Errors | Hallucinations result in incorrect output or prevent task completion. | Hallucinated outputs | 17 errors |
| | | Unspecified hallucination | 9 errors |
| | | Hallucination in reasoning | 5 errors |
| | | Hallucination in generated code | 3 errors |
| | | Hallucinated actions | 3 errors |
| | | Hallucinated assumptions | 3 errors |
| Output Errors | The agent generates output is missing, incorrect, or unexpected. | Low quality | 32 errors |
| | | Incorrect output | 32 errors |
| | | Incorrect output format or syntax | 23 errors |
| | | Incorrect output due to dependency issue | 10 errors |
| | | No output | 5 errors |
| | | Incorrect output despite correct plan execution | 4 errors |
| | | Incorrect classification | 4 errors |
| | | LLM judgments differ from human expectations | 3 errors |
| Code Errors | The agent generates unexecutable code or executable code that is incorrect. | Non-executable code | 33 errors |
| | | Wrong code result | 5 errors |

soning and goes directly into action (Caccia et al., 2024). Second, when an agent misses essential reasoning steps or actions required to successfully complete a task. These omissions may result from incorrect assumptions (Brown et al., 2024) or failure to explore and gather necessary information (Boisvert et al., 2024). Third, agents may struggle to plan over a long period of time (Murty et al., 2024; Gou et al., 2025), and fourth, they may construct a plan based on false assumptions (Wu et al., 2024a).

*Incorrect planning* errors are also evident in the characteristics of the plans. For instance, agents may generate non-actionable plans, which are overly complex to execute (Guo et al., 2024), incomplete (Shi et al., 2024b), or lack a complete description (Lal et al., 2024). In other instances, a plan does not match the stated goals, meaning that plan generated by an agent diverges from the intended task objectives or constraints because there is an incorrect condition analysis (Xu et al., 2024a), neglect of specific task constraints (Chen et al.), failure to adhere to prompt instructions (Jang et al., 2024), or unmet pre-conditions (Valmeekam et al., 2025; Hu et al., 2024). Lastly, in other instances, plans contained syntax errors (i.e., incorrect formatting) (Brown et al., 2024).

### 4.6. Non-Optimal Execution

*Non-optimal execution* errors refer to cases where agents' performance is inferior to what is expected, resulting in a decision-making process that is neither fully incorrect nor entirely successful. This type of error occurs when agents fail to validate the accuracy of their outputs (Pan et al., 2025), due to *"weaknesses in the agentic frameworks internal data-referencing or verification mechanisms"* (Islam et al., 2024b), or when an agent exceeds budgeted steps or execution time. Manifestations of *non-optimal execution* error include: (1) executing a task with inefficient perfor-

mance of the intermediate steps, (2) following an incorrect order of actions or logic in planning, (3) repeatedly performing ineffective actions, which in some instances lead to remaining stuck without making meaningful progress, and (4) performing unnecessary or redundant steps.

### 4.7. Incorrect Execution

*Incorrect execution* refers to errors in which agents are unable to successfully complete a task due to a wide variety of reasons. In some instances, the agents deliberately refuse to execute due to the lack of sufficient information (Sun et al., 2024; Wu et al., 2025), or simply fail to do so even when there is adequate information or reasoning (Wang et al., 2024b).

In other cases, *incorrect execution* stemmed from incorrect actions taken by agentic systems. Actions are an important part of agentic systems, as agentic systems typically use a model to select the actions and commonly act in external environments, such as APIs or webpages. Incorrect actions could occur when the agent interacted with external content, such as "assuming existence of files before checking" (Vergopoulos et al., 2025), attempting incorrect UI interactions such as swiping down to scroll up (Zhang et al., 2024), or attempting to take an invalid action that was not in the available action set (Liu et al., 2025).

Another cluster of reasons for *incorrect execution* relates to misalignment in plan execution, which manifests in:

1. Failure to complete plan execution, where an agent prematurely stops executing necessary actions to reach the desired goal. Premature termination occurred due to timeout (Mani & Attaranasl, 2025; Agashe et al., 2025), the agent's decision to finish the task before it is completed (Bai et al., 2024), stopping the execution of key tasks without a clear reason (Pan

et al., 2025), or *"stop[ping] solving the problem after encountering some difficulty"* (Yang et al., 2024).

2. Missing steps/actions in plan execution, where an agent omits required steps (Rivkin et al., 2024; Brown et al., 2024; Gou et al., 2025) or neglects processes during the task execution (Chu et al., 2024), even when the overall plan is correct.

3. Execution misalignment, which happens when agents' executed behavior does not align with logical reasoning (Pan et al., 2025; Boisvert et al., 2024).

Occasionally, *incorrect execution* errors arose from agents' insufficient communication. We identified two manifestations of this behavior. One was cross-agent communication failures, which refer to breakdowns in how agents exchange information (Lu et al., 2024) or consider input by other agents (Pan et al., 2025), or when they simply cannot learn from other agents (Tang et al., 2025). Second, some agents did not ask for clarification or request additional information, and instead proceeded to execute the task despite having incomplete inputs (de Oliveira et al., 2025; Pan et al., 2025). Lastly, agents may propagate errors, which consists of carrying forward and reinforcing an initial mistake or incorrect assumption across subsequent steps.

### 4.8. Tool Errors

*Tool errors* refer to errors related to the tools that agents must call and utilize to achieve their goals. Tools typically enable agentic systems to interact with external contexts in order to gather information or take an action in an external system. In some tool errors, the agent was unsuccessful in calling a tool because it used incorrect parameters, configuration, or syntax. *Tool errors* also arose when the tool that the agent attempted to call did not exist. In other cases, the agent called a tool that was incorrect for the desired action or not applicable for the current context, such as tools that do not have the functionality to accomplish the user's task (Winston & Just, 2025). In web-based tools, errors arose when the agent was unable to navigate websites correctly. For example, Bai et al. (2024) and Ou et al. (2024) describe cases when the agent attempted to click on un-clickable regions, and Lai et al. (2024) describes a case when a pop-up disrupted the agent's ability to interact with the website.

Another subset of tool errors stemmed from the tools themselves – for example, some tools threw errors such as failing to provide required information (Hu et al., 2023; Xiong et al., 2024; Tang et al.). When LLMs were used as a tool, their limitations in context length and input size could also cause errors.

### 4.9. Hallucination Errors

The propensity for LLMs to hallucinate – i.e., generate inaccurate or nonsensical outputs – extends to agentic systems. Researchers reported hallucinated outputs and generated code, such as non-existent events (Maharana et al., 2024) and function arguments (Bogin et al., 2024). Uniquely, agents also hallucinated actions, such as a nonsensical action (Jang et al., 2024) or actions that do not exist in a designated framework (Boisvert et al., 2024). They also hallucinated assumptions, such as incorrectly assuming that a previous action succeeded (Chen et al.), or reasoning, such as incorrectly understanding task progress when there has been no progress (Boisvert et al., 2024).

### 4.10. Output Errors

*Output errors* cover the ways that the agentic AI system's output is missing, incorrect, or unexpected. With over 100 entries in our taxonomy, *output errors* were one of the most apparent and commonly reported errors across the papers we surveyed. The majority of these errors focused on the correctness, simply reporting that the output was incorrect or missing. Some papers detailed how the output was not fully correct or not as expected. For example, low quality errors centered on quality issues like output completeness (Crispino et al., 2024; Xia et al., 2024; Hromei et al., 2024), ambiguous outputs (Islam et al., 2024b; Wu et al., 2025), grammatical mistakes (Hromei et al., 2024; Wu et al., 2025), and unnecessary repetition (Brown et al., 2024; Furuta et al., 2024).

### 4.11. Code Errors

*Code errors* include cases where the system was asked to generate executable code, but it failed to do so correctly. Code generation is commonly solved using agentic systems, especially for more complex coding problems, due to the need to locate relevant code and determine how to solve a coding problem. We found two main types of code errors: non-executable code, such as undefined variables (Guo et al., 2024) and wrong code result, i.e., "whether the code accomplishes the intended task and produces the correct output" (Winston & Just, 2025). Like *output problems*, *code problems* purely describe that an incorrect code output was produced without specifying underlying sources of error.

## 5. Discussion

We discuss the implications of our taxonomy, its practical application, future work, and limitations.

### 5.1. Implications

This work provides a comprehensive review of existing agentic AI literature, resulting in a taxonomy of errors in agentic systems, which fills a gap in existing work (Pan et al., 2025). Our work builds upon existing descriptions of failure modes in agentic (Zhan et al., 2023; Winston & Just, 2025; Pan et al., 2025; Sapkota et al., 2026) and AI systems by providing a systematic review of agentic AI papers and a taxonomy of the errors described. Most closely related to our work, Sapkota et al. list five main challenges in agentic systems: lack of causal understanding, inherited limitations from LLMs, incomplete agentic properties, limited long-horizon planning and recovery, and reliability and safety issues (Sapkota et al., 2026). Our error categories both confirm and expand upon these existing types, while finding gaps in the practical detection of errors. Several of our categories of errors primarily stem from the LLM itself, such as knowledge gaps, reasoning errors, or hallucinations. Others are more focused on the agentic qualities of the system, like issues in planning or executing actions or using tools incorrectly. Others may occur because of external systems or the system's interaction with external systems, like errors thrown by tools or issues with navigating website structures. Yet, we found fewer errors related to longer-term, downstream harms and very few errors in existing literature focused on the safety or responsibility of agentic systems for their users.

### 5.2. Proposed Use and Future Work

We expect that our taxonomy of agentic errors could be useful in the development of agentic frameworks and software, as well as in the evaluation and testing of agentic systems, similar to how Bagehorn et al. (2025)'s AI Risk Atlas has been used to identify benchmarks and mitigation strategies. Our taxonomy of agentic AI errors could also support agentic AI system development. With a formal taxonomy of the types of errors in agentic systems, frameworks can better support error handling. This is especially true in agentic systems, many of which have mechanisms for error detection and resolution (Bandi et al., 2025).

We also anticipate this taxonomy to be useful for testing and evaluation of agentic systems. One approach is by using the errors we identified in fault injection (Bandi et al., 2025). Creating benchmarks for agentic systems is more complex and challenging than prior AI systems, leading to benchmarks that don't appropriately capture performance (Zhu et al., 2025). Our taxonomy could be used to support benchmark creation by providing a set of errors that tasks should catch.

As innovations and technologies continue to evolve, the types and distributions of errors will change. Future work could continue to update and evolve this taxonomy based on research produced in the coming years. Future work could also investigate the frequencies of these error types, especially as researchers begin to include more comprehensive error analyses. Future reviews may want to expand beyond academic publications, taking into account a more diverse set of evaluations and reports about agentic systems.

Another area of future work is consideration of how these errors will be surfaced to users and how users might interact with these errors. Our taxonomy provides a foundation for thinking critically about the kinds of errors that occur in agentic systems and which may support human collaboration to enable resolution.

### 5.3. Limitations

Our paper is limited in several ways. While we tested and refined our search terms, it is possible that our searches missed some work that could have contributed to our taxonomy. Most papers did not include quantitative measures of the errors, so we were not able to provide insight into how often these errors occur in testing, only how often they are discussed in papers. Since there are no requirements for evaluation and discussion of all types of errors, our error counts do not necessarily reflect the commonality of errors, only how often researchers discuss these errors, and critically, only the ones that researchers chose to surface.

## 6. Conclusion

As agentic AI systems become more common, we need a better understanding of the kinds of errors that occur. In this paper, we presented a systematic review of papers describing agentic errors to report and classify the errors researchers have encountered in agentic systems. Our agentic AI error taxonomy defines 72 errors organized into 11 categories. This taxonomy provides a current and comprehensive understanding of agentic AI errors, enabling more systematic development and assessment of agents and moving us towards more proactive error handling approaches and ultimately more reliable agentic AI systems.

## 7. Impact Statement

This paper presents work with the goal of improving the design, development, and evaluation of agentic AI systems by identifying errors currently occurring in these kinds of systems. The errors and failures of agentic AI systems could have significant ethical and and societal impacts, especially as agentic AI systems interact more with the real world and fail in ways that can significantly impact people's lives. Our work does not address the ethics of these failures, but provides a taxonomy of errors which could be used for future ethical work around kinds of errors, responsibility for those errors, and impacts of those errors.

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

# A. Error Taxonomy Details

We provide more detail for each error category, including definitions for the errors within each category and the papers from which we identified each error, in Table 2. Note that some papers reported multiple different manifestations of the same error. In such instances, we only provided the citation of the paper once. Figure 1 presents the taxonomy visually.

| Error Subcategories: Definitions & Provenance | | |
|---|---|---|
| **Error** | **Definition** | **Papers** |
| **Instruction Understanding Errors** | | |
| LLM fails to understand instructions | The agent's incorrect behavior is attributed to a failure of its underlying LLM to understand instructions. Most often, these instructions are provided by a system or end-user prompt. | Akkil et al. (2024); Arabzadeh et al. (2024); Chen et al. (2024a; 2025b); Crispino et al. (2024); Deng et al. (2025); Fang et al. (2024); Goodell et al. (2025); Gou et al. (2025); He et al. (2024); Huang & Xiao (2024); Islam et al. (2024a); Jang et al. (2024); Jiang et al. (2024); Lu & Wang (2024); Moncada-Ramirez et al. (2025); Mou et al. (2024); Murty et al. (2024); Pan et al. (2025); Poglitsch et al. (2025); Press et al. (2024); Rivkin et al. (2024); Shi et al. (2024b;a); Wang et al. (2024a); Wu et al. (2025); Zhang et al. (2025c) |
| Prompt or instructions are ambiguous | The prompt given to the agent's underlying LLM lack clarity or can be interpreted in different ways. | Butala et al. (2024); Furuta et al. (2024); Kranti et al. (2024); Yu et al. (2025); Zhang et al. (2024) |
| LLM fails to understand task context | The agent's underlying LLM does not consider or is not given the additional context necessary to complete the task. | Huang et al. (2024c); Lai et al. (2024); Lu et al. (2025a) |
| **Input and Information Processing Errors** | | |
| Incorrect input retrieval | The agent selects the wrong input source or extracts the wrong information necessary to complete its task. | Boisvert et al. (2024); Cai et al. (2025); Chen et al. (2025a; 2024b); Gur et al. (2024); Hu et al. (2023); Jiao et al. (2024); Kim et al. (2024); Maharana et al. (2024); Press et al. (2024); Tang et al. (2025; 2024); Varela et al. (2023); Wang et al. (2024e;b); Xiong et al. (2024); Yang et al. (2024) |
| Visual input processing error | The agents misinterprets visual inputs due to visual input processing or understanding. | Awal et al. (2025); Gu et al.; Han et al. (2024); He et al. (2024); Huang et al. (2024a); Jang et al. (2024); Lai et al. (2024); Sravanthi et al. (2025); Wu et al. (2025); Xu et al. (2024a); Yu et al. (2025) |
| Problems with arithmetic and numbers | The agent incorrectly solves mathematical problems or incorrectly interprets numbers. | Awal et al. (2025); Chen et al. (2024b); Crispino et al. (2024); Goodell et al. (2025); Khandekar et al. (2024); Matlin et al. (2025b); Wu et al. (2024b; 2025) |
| Input formatting/syntax errors | The agent misinterprets inputs due to their format, characters, or syntax. | Brown et al. (2024); Chen et al. (2025a); He et al. (2024); Sravanthi et al. (2025); Wu et al. (2025) |
| Text comprehension error | The agent makes misinterprets input text including basic language misunderstanding, lexical problems, and difficulty with pronouns. | Fang et al. (2024); Gou et al. (2025); Huang & Xiao (2024); Kranti et al. (2024); Lei et al. (2025); Ouyang & Li (2023); Wang et al. (2024a) |
| LLMs don't understand 3D spaces | The agent's underlying LLM makes mistakes understanding three-dimensional spaces. | Chu et al. (2024); Kim et al. (2024); Kranti et al. (2024); Shi et al. (2024a) |
| **Knowledge Gaps** | | |

| | | |
|---|---|---|
| Lack of domain knowledge | The agent lacks domain-specific training to execute the task correctly. | Chen et al. (2024b; 2025a); Khandekar et al. (2024); Kranti et al. (2024); Lu et al. (2024; 2025a); Matlin et al. (2025b); Shi et al. (2024a); Tang et al. (2024); Wang et al. (2024b); Wu et al. (2025); Xu et al. (2024a) |
| Missing context | The agent has not built the correct context or is missing important context over the course of its task. | Butala et al. (2024); Chen et al.; Jang et al. (2024); Wang et al. (2024c); Xu et al. (2024a); Yang et al. (2024) |
| Context loss over time | The agent once had but lost the necessary context to complete the task. | AlRashed et al. (2025); Brown et al. (2024); Gou et al. (2025); Maharana et al. (2024); Murty et al. (2024); Pan et al. (2025); Wu et al. (2025); Zhou et al. (2024) |
| Lack of common sense | The agent lacks what papers described as "common sense:" common knowledge that dictates how the agent should behave in typical, everyday situations. | Chen et al. (2025b); Maharana et al. (2024); Rivkin et al. (2024); Tang et al. (2025); Wang et al. (2024a; 2025); Yu et al. (2025) |
| LLM training data affects correctness | The training data used to train agent's underlying LLM negatively affects how the agent performs its task. | AlRashed et al. (2025); Huang et al. (2024a); Matlin et al. (2025b); Ouyang & Li (2023) |
| **Reasoning Errors** | | |
| Unspecified reasoning errors | The agent incorrectly reasons about the task, but insufficient details explaining why are given. | Crispino et al. (2024); Duan et al. (2024); Tang et al. (2024; 2025); Wu et al. (2025); Xiong et al. (2024) |
| Information integration reasoning problems | The agent incorrectly reasoned about information integrated from multiple sources. | Awal et al. (2025); Huang & Xiao (2024); Huang et al. (2024a); Tang et al.; Wang et al. (2024d) |
| Problems generalizing or specializing knowledge | The agent either incorrectly generalizes from specific situations or incorrectly applies generalizations to specific situations. | AlRashed et al. (2025); Chen et al.; Huang et al. (2024b); Wu et al. (2025) |
| Causal reasoning errors | The agent incorrectly reasons about causation and cause and effect relationships. | Maharana et al. (2024); Matlin et al. (2025b); Wu et al. (2025) |
| Task complexity too high | The agent is unable to handle tasks with too many sources of information to reason about, or if there are too many steps needed to complete the task. | Sravanthi et al. (2025); Wu et al. (2025) |
| Training data biases reasoning | The agent performed above expectations due to its training data likely containing the data set being evaluated. | Matlin et al. (2025b) |
| **Incorrect Planning** | | |
| Plan does not match stated goals | The plan generated by the agent omits one or more requirements specified by the task. | Agashe et al. (2025); Arabzadeh et al. (2024); Chen et al. (2024b); Chen et al.; Gur et al. (2024); Hu et al. (2024); Huang et al. (2024b); Jang et al. (2024); Lal et al. (2024); Lei et al. (2025); Pan et al. (2025); Tang et al. (2025); Valmeekam et al. (2025); Wu & Henriksson (2024); Xu et al. (2024a) |

| | | |
|---|---|---|
| Missing steps/actions in plan | The plan generated by the agent is missing necessary steps to correctly complete execution. | Boisvert et al. (2024); Brown et al. (2024); Crispino et al. (2024); Hu et al. (2024); Lal et al. (2024); Lee et al. (2024); Rivkin et al. (2024); Tang et al. (2025); Wu et al. (2025) |
| Unspecified planning error | The plan generated by the agent is incorrect, but insufficient details explaining why are given. | Brown et al. (2024); Deng et al. (2025); Hu et al. (2023); Islam et al. (2024a); Tang et al. (2025); Wang et al. (2024e) |
| Unfollowable action plan | The plan generated by the agent is impossible to execute due to logical contradictions. | Guo et al. (2024); Gur et al. (2024); Huang et al. (2024c); Lal et al. (2024); Shi et al. (2024b) |
| Lack of a deliberate plan | The plan generated by the agent is not built on reasoned approaches, instead relying on memorized examples or greedy approaches. | Caccia et al. (2024); Cai et al. (2025); Chen et al.; Islam et al. (2024a) |
| Difficulty planning over a long time | The agent is unable to correctly plan due to the long-term nature of the task. | Gou et al. (2025); Murty et al. (2024) |
| False assumptions in plan | The plan generated by the agent is created with incorrect assumptions about the task. | Hu et al. (2024) |
| Syntax error in plan | The plan generated by the agent contains a syntax error. | Brown et al. (2024); Jiao et al. (2024) |
| **Non-optimal Execution** | | |
| Repetition in actions | The plan generated by the agent repeats actions unnecessarily. | Agashe et al. (2025); Boisvert et al. (2024); Brown et al. (2024); Grand et al. (2025); Huang et al. (2022; 2024c); Jang et al. (2024); Lee et al. (2024; 2025); Lo et al. (2023); Mou et al. (2024); Pan et al. (2025); Tang et al.; Yang et al. (2024); Zeng et al. (2024) |
| Incorrect steps/actions in plan | The plan generated by the agent includes steps or actions that are incorrect for the task. | Arabzadeh et al. (2024); Boisvert et al. (2024); Chen et al.; Chu et al. (2024); Hu et al. (2024); Jang et al. (2024); Jiao et al. (2024); Lal et al. (2024); Lee et al. (2024); Pan et al. (2025); Shi et al. (2024b); Wang et al. (2024d) |
| Action budget limits | The agent has exceeded the budget provided by the service that provides one or more agent capabilities. | Brown et al. (2024); He et al. (2024); Huang et al. (2022); Jiang et al. (2024); Liu et al. (2025); Press et al. (2024); Shao et al. (2024); Wang et al. (2025); Winston & Just (2025); Yang et al. (2024) |
| Unspecified non-optimal execution errors | The agent completes its task, but is inefficient in doing so. Insufficient details are explaining why are given. | Bai et al. (2024); Cai et al. (2025); Chen et al.; Krishnamurthy et al. (2024); Liu et al.; Pan et al. (2025); Varela et al. (2023) |
| Inadequate verification | The agent fails to validate or verify outputs, actions, or outcomes, leading to incorrect task completion. | Cai et al. (2025); Islam et al. (2024b); Pan et al. (2025) |
| Extra actions | The plan generated by the agent includes unnecessary extra actions that are not required to complete the task. | Pan et al. (2025) |
| **Incorrect Execution** | | |

| | | |
|---|---|---|
| Unspecified incorrect actions | The agent performs an incorrect action for the task, but insufficient details explaining why are given. | Agashe et al. (2025); Arabzadeh et al. (2024); Bai et al. (2024); Brown et al. (2024); Chen et al. (2024a;b); Crispino et al. (2024); Dong et al. (2024); Grand et al. (2025); Hoefer et al. (2025); Huang et al. (2024b); Jang et al. (2024); Jiao et al. (2024); Kranti et al. (2024); Lin et al. (2025); Mandi et al. (2024); Ou et al. (2024); Pan et al. (2025); Wang et al. (2025); Wu & Henriksson (2024); Yu et al. (2025); Zhang & Lee (2024) |
| Failure to complete plan execution | The agent does not execute all the steps in its plan. | Agashe et al. (2025); Bai et al. (2024); Fang et al. (2024); He et al. (2024); Huang et al. (2022; 2024b); Kim et al. (2024); Lim et al. (2024); Liu et al.; Mani & Attaranasl (2025); Pan et al. (2025); Valmeekam et al. (2025); Yang et al. (2024); Zhou et al. (2024) |
| Invalid actions | The agent attempts to perform an action that is not applicable to user interface element. | Liu et al. (2025); Lo et al. (2023); Ou et al. (2024); Wang et al. (2024c); Xu et al. (2024b); Zeng et al. (2024) |
| Missing steps/actions in plan execution | The agent omits one or more steps from its plan when executing its task. | Bogin et al. (2024); Brown et al. (2024); Chu et al. (2024); Gou et al. (2025); Press et al. (2024); Rivkin et al. (2024); Shi et al. (2024a); Yu et al. (2025); Zhou et al. (2024) |
| Incorrect UI interaction | The agent interacts with the wrong element or incorrectly navigates a user interface. | Chen et al.; Lee et al. (2024); Murty et al. (2024); Ouyang & Li (2023); Wang et al. (2024c); Yu et al. (2025); Zhang et al. (2024) |
| Cross-agent communication failures | The agent does not adequately share or consider information shared by and shared with other agents. | Lu et al. (2024); Pan et al. (2025); Tang et al. (2025); Wu et al. (2024a) |
| Propagation of errors | The agent's execution is disrupted by errors that occurred in an earlier step of its task execution. | Bai et al. (2024); Bogin et al. (2024); Gu et al.; Guo et al. (2024); Huang et al. (2022); Jang et al. (2024); Mandi et al. (2024); Pan et al. (2025); Shi et al. (2024b); Tang et al.; Wang et al. (2025) |
| Unspecified execution errors | The agent failed to complete the task during execution, but insufficient details explaining why are given. | Hu et al. (2024); Tang et al. (2025); Wu et al. (2024b); Xu et al. (2024b); Yang et al. (2024) |
| Agent refusal to execute | The agent refuses to do some or all of the task. | Sun et al. (2024); Wu et al. (2025); Zeng et al. (2024) |
| Executing non-existent actions | The agent attempts to execute actions or call APIs that don't exist. | Agashe et al. (2025); Boisvert et al. (2024); Pan et al. (2025) |
| False assumptions in execution | The agent makes incorrect assumptions when executing its task. | Brown et al. (2024); Wu et al. (2025) |
| Doesn't ask for clarification | The agent does not request additional information when facing unclear situations or incomplete data. | de Oliveira et al. (2025); Pan et al. (2025) |

**Tool Errors**

| Unable to navigate websites | The agent's incorrectly navigates a web page. | Agashe et al. (2025); Akkil et al. (2024); Awal et al. (2025); Bai et al. (2024); Furuta et al. (2024); He et al. (2024); Lai et al. (2024); Lee et al. (2025); Ou et al. (2024); Vergopoulos et al. (2025); Yu et al. (2025) |
|---|---|---|
| Tool is incorrect for the desired action | The agent's selected tool is incorrect for one or more actions in its task. | Chen et al.; Lin et al. (2025); Ouyang & Li (2023); Rivkin et al. (2024); Shi et al. (2024b); Wang et al. (2024a); Winston & Just (2025); Yang (2024); Zhang et al. (2025c) |
| Tool is called with incorrect parameters | The agent's selected tool, API, or service is called with incorrect parameters. | Bogin et al. (2024); Jiao et al. (2024); Lin et al. (2025); Ouyang & Li (2023); Rivkin et al. (2024); Tang et al.; Wang et al. (2024a); Winston & Just (2025); Zhang et al. (2025c) |
| LLM context length and input size limitations | The agent has exceeded an LLMs service's context length or input limitations. | Brown et al. (2024); Huang & Xiao (2024); Lei et al. (2025); Lo et al. (2023); Press et al. (2024); Rivkin et al. (2024); Shao et al. (2024); Shi et al. (2024b); Winston & Just (2025) |
| Tool throws an error | The agent's selected tool throws an error or exception that the agent is unable to handle. | Fuchs et al. (2023); Hu et al. (2023); Jain et al. (2024); Rivkin et al. (2024); Tang et al.; Winston & Just (2025); Xiong et al. (2024) |
| Tool not applicable for current context | The agent has selected the incorrect tool, API, or service given the current context of the task. | Huang et al. (2022); Ouyang & Li (2023); Tang et al.; Winston & Just (2025) |
| Tool is called with incorrect syntax | The agent's selected tool tool, API, or service is called with incorrect syntax. | Gou et al. (2025); Ouyang & Li (2023); Xiong et al. (2024); Zhang & Lee (2024); Zhang et al. (2025c) |
| Other types of tool errors | The agent's selected tool is incorrectly used, but insufficient details why are given. | Lim et al. (2024); Rivkin et al. (2024); Shao et al. (2024); Shi et al. (2024b) |
| Tool is called with incorrect configuration | The agent's selected tool, API, or service is called with incorrect configuration or specifications. | Jiao et al. (2024); Tang et al. |
| Tool does not exist | The agent's selected tool does not exist or is unavailable to the agent. | Liu et al. (2025); Zeng et al. (2024) |
| **Hallucination Errors** | | |
| Hallucinated outputs | The agent hallucinates when generating output. | Crispino et al. (2024); Deng et al. (2025); Huang & Xiao (2024); Islam et al. (2024b); Lu et al. (2025a); Maharana et al. (2024); Rivkin et al. (2024); Tang et al.; Wang et al. (2024b; 2025); Watson et al. (2025); Wu et al. (2025); Xiong et al. (2024) |
| Unspecified hallucination errors | The agent hallucinates somewhere in its task execution, but insufficient details why are given. | Duan et al. (2024); Gou et al. (2025); Jiang et al. (2024); Lai et al. (2024); Lo et al. (2023); Moncada-Ramirez et al. (2025); Sun et al. (2024); Zhao et al. (2024) |
| Hallucination in reasoning | The agent hallucinates while reasoning during its task. | Boisvert et al. (2024); Goodell et al. (2025); He et al. (2024); Moncada-Ramirez et al. (2025) |

| | | |
|---|---|---|
| Hallucination in generated code | The agent hallucinates when generating code. | Bogin et al. (2024); Guo et al. (2024); Yang (2024) |
| Hallucinated actions | The agent hallucinates actions during planning or execution. | Boisvert et al. (2024); Jang et al. (2024); Winston & Just (2025) |
| Hallucinated assumptions | The agent hallucinates assumptions during planning or execution. | Chen et al.; Huang et al. (2024c); Poglitsch et al. (2025) |
| **Output Errors** | | |
| Low quality | The agent's output is lower-than-expected quality. This includes language errors (misspellings, incorrect grammar, and incorrect syntax), inconsistency in responses, ambiguous responses, correct but incomplete output, and repeated outputs. | Akkil et al. (2024); Brown et al. (2024); Crispino et al. (2024); Dong et al. (2024); Furuta et al. (2024); Hromei et al. (2024); Islam et al. (2024b); Jiang et al. (2024); Liu et al. (2025); Matlin et al. (2025b); Murty et al. (2024); Ou et al. (2024); PP & Iyer (2024); Shea & Yu (2024); Tang et al. (2024); Tang et al.; Winston & Just (2025); Wu et al. (2025); Xia et al. (2024) |
| Incorrect output | The agent's output is incorrect for the given task. | Butala et al. (2024); Crispino et al. (2024); Dong et al. (2024); Gou et al. (2025); Hoefer et al. (2025); Huang et al. (2022); Islam et al. (2024b); Jain et al. (2024); Khandekar et al. (2024); Kim et al. (2024); Liu et al.; Lu & Wang (2024); Lu et al. (2025a); Maharana et al. (2024); Matlin et al. (2025b); PP & Iyer (2024); Press et al. (2024); Shao et al. (2024); Varela et al. (2023); Wang et al. (2024e); Winston & Just (2025); Wu & Henriksson (2024); Wu et al. (2025); Xiong et al. (2024); Yang et al. (2024); Zhang & Lee (2024) |
| Incorrect output format or syntax | The agent's output has incorrect syntax or incorrect data format. | Chen et al.; Crispino et al. (2024); Fang et al. (2024); Gou et al. (2025); Huang & Xiao (2024); Huang et al. (2024c); Jain et al. (2024); Jiang et al. (2024); Matlin et al. (2025b); Poglitsch et al. (2025); PP & Iyer (2024); Rivkin et al. (2024); Sun et al. (2024); Tang et al. (2025); Wu et al. (2025); Xia et al. (2024); Yang (2024); Yu et al. (2025); Zhang et al. (2025a) |
| Incorrect output due to dependency issue | The agent's output is incorrect due to one or more data dependencies. | Fang et al. (2024); Gou et al. (2025); Huang & Xiao (2024); Mani & Attaranasl (2025); Matlin et al. (2025b); Tang et al.; Zhang et al. (2025c) |
| No output | The agent does not produce any output. | Crispino et al. (2024); Gandhi et al. (2024); He et al. (2024); Hoefer et al. (2025); Shao et al. (2024) |
| Incorrect output despite correct plan execution | The agent's output is incorrect, even though the agent fully followed a correct plan for the task. | Goodell et al. (2025); Liu et al. (2025); Yang et al. (2024) |
| Incorrect classification | The agent misclassifies one or more inputs during a classification task. | Cheng et al. (2024); Hromei et al. (2024); Lu et al. (2025a); Xu et al. (2024a) |
| LLM judgments differ from human expectations | The agent's output and classifications differ when compared to a group of human evaluators doing the same task. | Kumar et al. (2025) |

**Code Errors**

| | | |
|---|---|---|
| Non-executable code | The agent generates code that does not execute. | Bogin et al. (2024); Dong et al. (2024); Gandhi et al. (2024); Gou et al. (2025); Grand et al. (2025); Guo et al. (2024); Islam et al. (2024a); Jain et al. (2024); Lei et al. (2025); Rivkin et al. (2024); Tang et al.; Winston & Just (2025); Yang (2024); Zhang et al. (2025b) |
| Wrong code result | The agent generates executable code, but it produces the wrong result. | Dong et al. (2024); Goodell et al. (2025); Lim et al. (2024); PP & Iyer (2024); Winston & Just (2025) |

*Table 3.* Papers from which errors were extracted. Here we provide definitions of each of the errors and the source paper for that error. Papers that had multiple errors in the same category or only cited once.

[1]Anonymous Institution, Anonymous City, Anonymous Region, Anonymous Country. Correspondence to: Anonymous Author <anon.email@domain.com>.

Preliminary work. Under review by the International Conference on Machine Learning (ICML). Do not distribute.

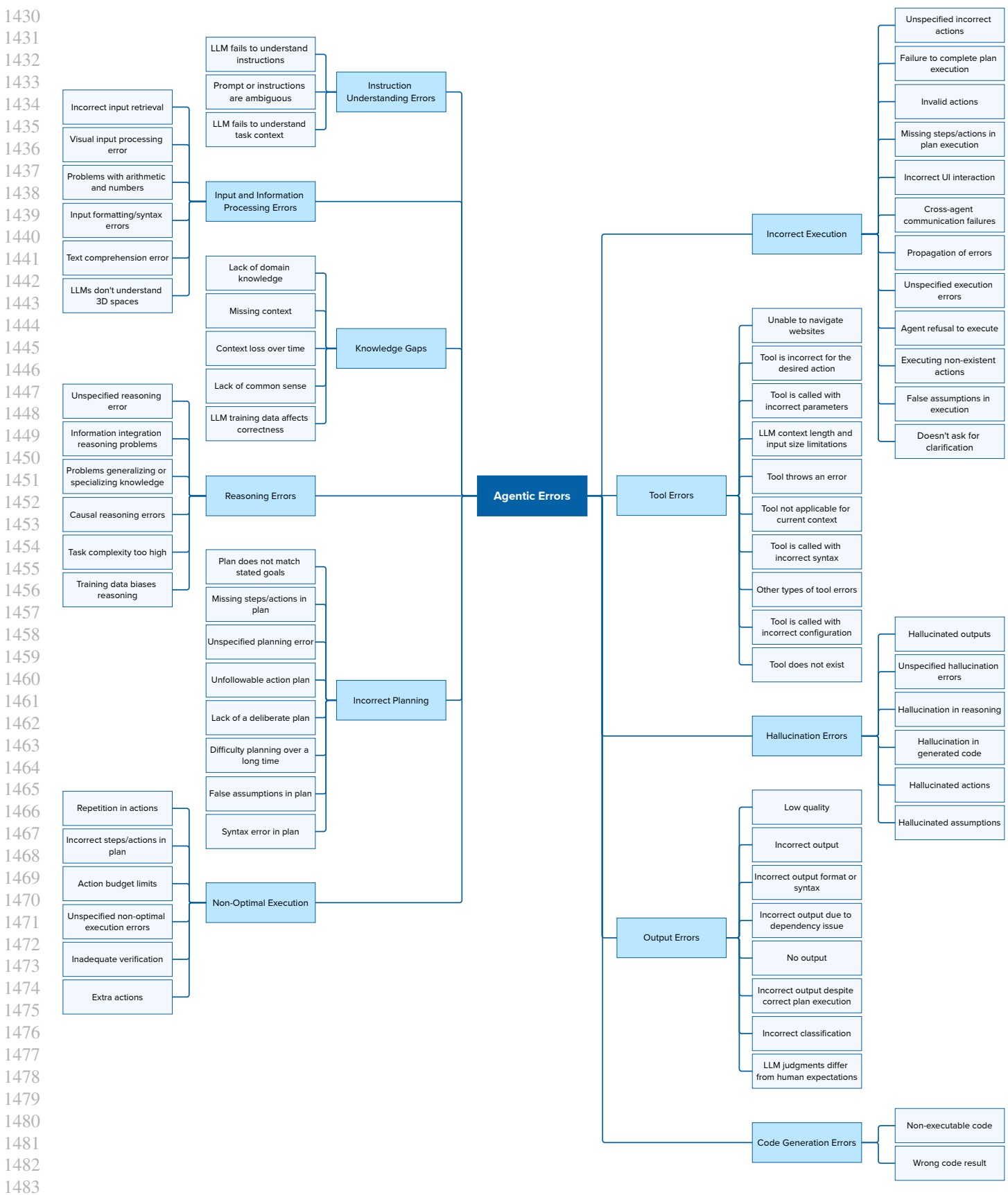

*Figure 1.* Taxonomy