# OpenReview forum: "A Taxonomy of Agentic Errors: A systematic review of how agentic AI systems can fail"
_ICML.cc/2026/Conference — Submitted to ICML 2026_

### Official Review · Reviewer_UM1Z · 2026-03-06

**Soundness:** 2
**Presentation:** 2
**Significance:** 2
**Originality:** 1
**Overall Recommendation:** 2
**Confidence:** 4

**Summary:**

In my understanding, ICML submission policy requires original research to be submitted, and this is a survey / literature review article. The author's main contribution is a taxonomy of agentic AI errors, scraped from 1376 research papers. Because this is lacking in novel original research, I recommend rejection for being out of scope for this venue.

**Compliance With Llm Reviewing Policy:**

Affirmed.

**Final Justification:**

I am awaiting feedback from the AC to determine if my reasons for rejection due to being out of scope for the conference are valid.

**Key Questions For Authors:**

see above

**Limitations:**

see above

**Strengths And Weaknesses:**

see above

---

> ### Author Rebuttal · Authors · 2026-03-30
>
> We thank the reviewer for their feedback. In what follows, we explain how our research fits into the call of the conference and explain how taxonomies as research artifacts are valuable.
>
> We contribute an original analysis of errors in agentic evaluations in the form of a taxonomy, which we developed through rigorous analysis using the PRISMA method. We chose to use a systematic literature review intentionally to consolidate fragmented scholarship around agentic errors and provide original findings that demonstrate improved understanding of agentic errors. Our paper falls under the category of “evaluation” from the call for papers, which includes “**meta studies**”. Further, the reviewer instructions specify that “originality does not necessarily require introducing an entirely new method. Rather, a work that provides **novel insights by evaluating existing methods**, or **demonstrates improved understanding** is also equally valuable.” The reviewing instructions also state that “originality may arise from **creative combinations of existing ideas**, application to a real-world use case, or removing restrictive assumptions from prior theoretical results.”
>
> We chose to submit this work to ICML, as we found that ICML was the primary venue for research around agentic systems and their errors. Papers with similar contributions to ours have been published, such as:
>
> Liao, T., Taori, R., Raji, I. D., & Schmidt, L. (2021, January). Are we learning yet? a meta review of evaluation failures across machine learning. In Thirty-fifth Conference on Neural Information Processing Systems Datasets and Benchmarks Track (Round 2).
>
> Atapattu, T., Thilakaratne, M., Do, D. N., Herath, M., & Falkner, K. E. (2025). Exploring the Role of Mental Health Conversational Agents in Training Medical Students and Professionals: A Systematic Literature Review. Findings of the Association for Computational Linguistics: ACL 2025, 20785-20798.
>
> Sen, I., Lutz, M., Rogers, E., Garcia, D., & Strohmaier, M. (2025). Missing the margins: A systematic literature review on the demographic representativeness of LLMs. Findings of the Association for Computational Linguistics: ACL 2025, 24263-24289.
>
> In addition to the fact that review papers are explicitly recognized as relevant contributions at this conference, we emphasize that the development of taxonomies constitutes foundational research in its own right.
>
> First, taxonomies enable cumulative and actionable research by providing shared structures that prevent each new study from “reinventing” risk categories. A notable example is the AI Risk Repository, which aggregates and organizes over 1,700 risks into a living, extensible framework that can be continuously updated and reused by the community. Similarly, our taxonomy provides a foundation for iterative refinement as new agentic capabilities emerge and novel failure modes are discovered, enabling the research community to collectively advance the understanding of agent behavior.
>
> Second, taxonomies provide a shared language to support communication across domains. Previous research focused on AI Safety has drawn attention to the conceptual fragmentation that emerges when different research communities use inconsistent terminology. The taxonomy we propose in this work establishes shared terminology that enables practitioners, researchers, and policymakers to identify, discuss, and collectively address agentic AI failures using consistent categories rather than ad-hoc descriptions.
> Lastly, taxonomies provide systematic coverage and surface blind spots. For agentic AI systems in particular, without systematic error categorization, development teams most likely will tend to address only the most visible failures while missing systematic failure modes that emerge from agent architecture and planning constraints. Thus, taxonomies like the one we describe in this article can serve as groundwork for assessing and prioritizing different categories of systemic risk and for developing targeted responses.

---

> > ### Author Rebuttal · Reviewer_UM1Z · 2026-03-31
> >
> > This was a difficult paper to review because I could not find precedent for similar papers at ICML (the authors give some examples, but these are for other venues and different conference tracks). The authors claim the conference explicitly permits "review papers". I cant see clear evidence for this in the call for papers, or in papers accepted from previous years. The call for papers permits "evaluations (meta-studies)", but it isnt clear what this means; if it extends to papers summarising the findings of many other papers (e.g. this agentic failure taxonomy), which could skew more towards a review article, or if it requires some more complex synthesis and statistical analysis of the findings of other papers, to e.g. prove a hypothesis, such as in a meta-analysis of clinical trials. As this was a sub-category of "evaluations" (and not review articles), I assumed the latter, setting the bar that the meta-study should test some hypothesis (e.g. meta-analysis of many RCTs with conflicting results establishes a drug is ineffective). I didnt find the paper satisfied this criteria.
> >
> > Prior to the discussion period, I requested feedback from the AC to determine if my recommendation to reject was in line with the reviewer guidelines or not. I am still waiting for a reply, and will raise my score if the AC determines that meta reviews such as these are permitted to the main conference track. I have selected `partially resolved' because I do not believe the work needs significant updates as it is.

---

### Official Review · Reviewer_mQ5Q · 2026-03-12

**Soundness:** 3
**Presentation:** 3
**Significance:** 3
**Originality:** 2
**Overall Recommendation:** 3
**Confidence:** 4

**Summary:**

This paper presents a Taxonomy of Agentic Errors, a structured classification of failure modes in emerging agentic AI systems. To systematically understand failures of AI agents, the authors conduct a systematic literature review following the PRISMA methodology. The study initially identifies 1,379 relevant publications, from which 123 papers describing agentic failures are selected. The paper provides a comprehensive, structured overview of agentic AI failure modes, offering a shared vocabulary for researchers and practitioners studying the reliability and safety of autonomous AI agents.
Although the paper does not introduce a novel algorithm or modeling technique, it performs an important conceptual groundwork for the rapidly evolving field of agentic AI. The proposed taxonomy is extensive, clearly structured, and highly relevant to ongoing efforts in agent evaluation and safety. By consolidating fragmented observations from the literature into a coherent framework, the paper provides a valuable reference for researchers studying agent reliability, debugging, and evaluation.

**Compliance With Llm Reviewing Policy:**

Affirmed.

**Key Questions For Authors:**

1. In many cases, agent failures occur as cascading processes, where an early-stage error (e.g., instruction misunderstanding) leads to planning errors and eventually incorrect execution. Beyond the current flat taxonomy of 11 categories, how might the authors model dependencies or propagation paths between error types?
2. Errors mentioned in academic literature do not necessarily correspond to the most frequent or critical failures in real-world systems.
Do the authors envision developing metrics or evaluation frameworks that could quantify the severity, frequency, or risk level of each error category when applied to real agentic systems?

**Limitations:**

-

**Strengths And Weaknesses:**

Strengths:

1. Timely and Comprehensive Meta-Analysis: This paper provides a systematic and large-scale synthesis of prior research, aggregating fragmented observations from the literature into a unified taxonomy. The scale of the analysis—spanning 652 documented errors across 123 papers—makes this work a valuable foundational reference for the field.
2. Granular and Agent-Specific Error Categorization: The taxonomy goes beyond conventional LLM failure categories and captures errors unique to autonomous agent systems. In this sense, the work provides an important conceptual baseline for future empirical research on agent robustness and safety.

Weaknesses & Limitations:

1. As acknowledged by the authors, the counts reported in the paper reflect how often errors were mentioned in the literature, rather than how frequently they occur in practice. The taxonomy does not capture real-world error frequency, severity of each failure type, or relative importance of different error categories.
2. A hierarchical or causal modeling framework could improve the explanatory power of the taxonomy.
Some categories in the taxonomy correspond to underlying causes (e.g., knowledge gaps, hallucinations), while others describe observable outcomes (e.g., incorrect execution, output errors, code failures). However, failures in agentic systems often occur as multi-stage cascades: instruction misunderstanding → planning failure → incorrect execution.
3. The current taxonomy presents these categories in a mostly parallel structure, which limits its ability to capture causal relationships between error types.
4. The dataset used in the study consists exclusively of peer-reviewed academic papers. However, many real-world failures of agentic systems occur in real-world environments such as industrial deployments, production AI systems, and open-source agent frameworks.

---

> ### Author Rebuttal · Authors · 2026-03-30
>
> Thank you for your review. Your insightful questions identified some areas where our discussion could use additional clarity. We plan on updating the discussion to address these points as below.
>
> We agree that agent failures often occur as cascading processes. Many of the errors in the taxonomy are causal; they are the reason that the agent has incorrect output or incorrect behavior. However, the papers themselves rarely analyzed potential chains of failures beyond the initial causes. Identifying causal relationships between errors was further limited by the differences in how each paper identified, detailed, and analyzed each error. Those differences made it not possible to identify causal patterns. We will acknowledge the cascading nature of failures and explain why we were unable to include it in the taxonomy more explicitly.
>
> We agree that it is necessary to start establishing and understanding the causal relationships and cascades of failures noted in your question. A taxonomy like this one is the first step towards that understanding. This taxonomy of errors provides a shared language and foundation for future research to build upon. Future descriptions or analyses of errors, whether in research or in the real-world, can begin to fill in those gaps by using our taxonomy.
>
> We agree with the reviewer that there is often a disconnect between academia and the real-world. We would have liked to extend this analysis to real-world use cases, but publicly available information about how real-world agents fail do not provide the same level of detailed error analysis as those in the research community. Thus, we started with what was readily available.
>
> Evaluation frameworks could potentially be extended to quantify a subset of the errors in our taxonomy, but not all. For example, identifying and quantifying errors like tool exceptions, an agent gathering the wrong data, or an agent not completing its task could be done within existing agentic AI frameworks. Other types of errors are more challenging to quantify. Determining if the agent is taking the most optimal sequence of actions or providing a full response is more difficult to measure with our current tools.
>
> Quantifying the risk per error is more difficult since it is use-case dependent. The severity of the same error can vary dramatically across different scenarios. For example, a banking agent that can modify someone’s financial information is riskier than a creative writing agent that is modifying a story. The type of error alone cannot define the risk. We believe a taxonomy can provide guidance for such directions. A person interested in understanding the risk of their agentic system can now refer to a taxonomy to see which errors may be relevant to them, map them to their use case and make an assessment of how severe the consequences of that error might be. Without such a taxonomy, the consideration of errors, and thus, risks, may not be exhaustive.

---

> > ### Author Rebuttal · Reviewer_mQ5Q · 2026-04-05
> >
> > Thank you for the thoughtful rebuttal. I appreciate the clarifications and agree that the taxonomy can serve as a useful first step toward a more systematic understanding of agentic AI failures.
> > That said, my main concerns remain only partially addressed. The taxonomy remains primarily descriptive rather than explanatory, particularly regarding failure cascades and causal dependencies across error types. In addition, the reliance on peer-reviewed academic papers limits the connection to real-world agent failures and practical risk.
> >
> > More broadly, while it is not entirely clear to me whether ICML explicitly welcomes review-style papers of this kind, I do recognize the value of this submission as a review-oriented and conceptual contribution. However, this does not change my evaluation relative to the conference bar.
> > I acknowledge the value of the work as a review paper, but I am unable to raise my score.

---

### Official Review · Reviewer_zi75 · 2026-03-13

**Soundness:** 3
**Presentation:** 3
**Significance:** 3
**Originality:** 2
**Overall Recommendation:** 5
**Confidence:** 3

**Summary:**

This paper provides a taxonomy of agentic AI system failures, based on a systematic review of agentic AI research. They find 652 examples of agentic AI errors, and provide a helpful set of what appear to be mutually exclusive and jointly exhaustive categories.

**Compliance With Llm Reviewing Policy:**

Affirmed.

**Key Questions For Authors:**

None

**Limitations:**

Yes

**Strengths And Weaknesses:**

Soundness:
- The use of PRISMA for systematic review is good
- The search terms made sense, but could have been expanded. A purposive sample of papers could have been studied to see if there are any other synonyms for 'error' to add to expand the search terms.
- This approach aims to synthesise what the research community determines constitutes an error. This is commendable, because it is bottom-up and inductive. However, it could also result in a taxonomy which is suboptimal, because the papers being drawn from might be less rigorous in their definition and application of terms like 'error'.
- The definition of 'non-optimal execution' seems a bit narrower than the name would imply; it only describes 'extra, unnecessary actions', but is that the only way for execution to be non-optimal?
- Overall, the taxonomy of errors seems sound; each category feels mutually exclusive, and they seem likely to be jointly exhaustive for the vast majority of errors.

Presentation:
- Overall the paper is well written and well structured.
- In a few places the wording could have been better:
  - "the taxonomy we propose is comprehensive" - this should probably say 'aims to be comprehensive'
  - The definitions of Knowledge Gaps and Reasoning Errors is the same ('the agent makes mistakes due to its knowledge or its task approach'). I assume this is an error.

Significance:
- This is an important class of problems, and the paper advances our understanding of them.

Originality:
- The work is fairly original; while previous papers have attempted similar taxonomies, none have attempted to be as comprehensive as this. Leveraging existing literature to source examples is an original approach.

---

> ### Author Rebuttal · Authors · 2026-03-30
>
> Thank you to the reviewer. We appreciate the positive review and the feedback.
>
> We will clarify the wording as well as the definition of non-optimal execution. We did experiment with the search terms and found that error and failure typically covered the kinds of issues we were looking for. We will clarify our methods to include this.

---

> > ### Author Rebuttal · Reviewer_zi75 · 2026-04-03
> >
> > The suggestion modification will improve the paper. I keep my original scores which were already positive.

---

### Official Review · Reviewer_Cp7Y · 2026-03-26

**Soundness:** 1
**Presentation:** 1
**Significance:** 1
**Originality:** 1
**Overall Recommendation:** 2
**Confidence:** 4

**Summary:**

This paper is a review-style article which identifies and organizes a range of errors that AI systems make.

**Compliance With Llm Reviewing Policy:**

Affirmed.

**Key Questions For Authors:**

As this type of paper is usually not a good fit for this venue, I am inclined to leave this section blank.

**Limitations:**

N/A.

**Strengths And Weaknesses:**

This paper is an unusual submission to ICML in that it is a review paper without new quantitative (technical track) or qualitative (position paper track) contributions, beyond the identification and organization of error types as defined by the authors. While having such a collection is certainly a good thing, to my knowledge this is not the type of paper that ICML solicits or publishes.

---

> ### Author Rebuttal · Authors · 2026-03-30
>
> We thank the reviewer for their feedback. In what follows, we explain how our research fits into the call of the conference and explain how taxonomies as research artifacts are valuable.
>
> We contribute an original analysis of errors in agentic evaluations in the form of a taxonomy, which we developed through rigorous analysis using the PRISMA method. We chose to use a systematic literature review intentionally to consolidate fragmented scholarship around agentic errors and provide original findings that demonstrate improved understanding of agentic errors. Our paper falls under the category of “evaluation” from the call for papers, which includes “**meta studies**”. Further, the reviewer instructions specify that “originality does not necessarily require introducing an entirely new method. Rather, a work that provides **novel insights by evaluating existing methods**, or **demonstrates improved understanding** is also equally valuable.” The reviewing instructions also state that “originality may arise from **creative combinations of existing ideas**, application to a real-world use case, or removing restrictive assumptions from prior theoretical results.”
>
> We chose to submit this work to ICML, as we found that ICML was the primary venue for research around agentic systems and their errors. Papers with similar contributions to ours have been published, such as:
>
> Liao, T., Taori, R., Raji, I. D., & Schmidt, L. (2021, January). Are we learning yet? a meta review of evaluation failures across machine learning. In Thirty-fifth Conference on Neural Information Processing Systems Datasets and Benchmarks Track (Round 2).
>
> Atapattu, T., Thilakaratne, M., Do, D. N., Herath, M., & Falkner, K. E. (2025). Exploring the Role of Mental Health Conversational Agents in Training Medical Students and Professionals: A Systematic Literature Review. Findings of the Association for Computational Linguistics: ACL 2025, 20785-20798.
>
> Sen, I., Lutz, M., Rogers, E., Garcia, D., & Strohmaier, M. (2025). Missing the margins: A systematic literature review on the demographic representativeness of LLMs. Findings of the Association for Computational Linguistics: ACL 2025, 24263-24289.
>
> In addition to the fact that review papers are explicitly recognized as relevant contributions at this conference, we emphasize that the development of taxonomies constitutes foundational research in its own right.
>
> First, taxonomies enable cumulative and actionable research by providing shared structures that prevent each new study from “reinventing” risk categories. A notable example is the AI Risk Repository, which aggregates and organizes over 1,700 risks into a living, extensible framework that can be continuously updated and reused by the community. Similarly, our taxonomy provides a foundation for iterative refinement as new agentic capabilities emerge and novel failure modes are discovered, enabling the research community to collectively advance the understanding of agent behavior.
>
> Second, taxonomies provide a shared language to support communication across domains. Previous research focused on AI Safety has drawn attention to the conceptual fragmentation that emerges when different research communities use inconsistent terminology. The taxonomy we propose in this work establishes shared terminology that enables practitioners, researchers, and policymakers to identify, discuss, and collectively address agentic AI failures using consistent categories rather than ad-hoc descriptions.
> Lastly, taxonomies provide systematic coverage and surface blind spots. For agentic AI systems in particular, without systematic error categorization, development teams most likely will tend to address only the most visible failures while missing systematic failure modes that emerge from agent architecture and planning constraints. Thus, taxonomies like the one we describe in this article can serve as groundwork for assessing and prioritizing different categories of systemic risk and for developing targeted responses.

---

> > ### Author Rebuttal · Reviewer_Cp7Y · 2026-03-31
> >
> > Thank you to the authors for their response. While I agree that review papers and taxonomies are useful, the authors will notice that the papers they cited are submitted to different conferences (ACL) and tracks (Datasets and Benchmarks at NeurIPS). To meet the bar for a strong ICML submission to this main conference track, there unfortunately needs to be significantly more work done.

---

### Decision · Program_Chairs · 2026-04-30

**Decision:**

Reject

**Comment:**

Reviewers agreed that the paper is well-written and well-executed and the taxonomy offered could be a valuable first step toward understanding agentic errors. However, they thought the proposed taxonomy has limited explanatory power (e.g., regarding the cascading nature of agentic errors), is largely descriptive of the literature, and is focused narrowly on errors identified in research papers (as opposed to real-world instances). For those reasons, they questioned whether the paper hits the bar for a main-track ICML paper.